# Energy Accumulation Law of Different Forms of Coal–Rock Combinations

Zibo Li [1], Guohua Zhang [2,*], Yubo Li [1], Wenjun Zhou [2], Tao Qin [2], Li Zeng [2] and Gang Liu [2]

[1] School of Safety Engineering, Heilongjiang University of Science and Technology, Harbin 150022, China
[2] School of Mining Engineering, Heilongjiang University of Science and Technology, Harbin 150022, China; 18944630110@163.com (G.L.)
* Correspondence: 1994800303@usth.edu.cn

**Abstract:** Coal–rock disasters are becoming more and more severe as the intensity of coal mining increases. Due to its destructive power and resulting extensive area damage, rock burst is among the most critical threats to coal mine safety. It results from the combined action of the coal and the rock when affected by the mining process. To this end, we used a combination of coal and rock to conduct our studies. Combining a uniaxial compression experiment with theoretical analysis, this work investigated how different lithologies and coal–rock height ratios affect the mechanical properties of this combination and the law governing energy accumulation. We determined the following: When the coal–rock height ratios are dissimilar, the peak strength and modulus of elasticity of the combination show a negative correlation with the coal thickness share, and the pre-peak energy accumulation and impact energy index of the combination is positively correlated with the coal thickness percentage. In combination with the same coal–rock height ratio, the peak strength, elastic modulus, pre-peak energy accumulation, and impact energy index all increase with increased rock strength and elastic modulus. The presence of a hard rock layer affects the accumulation of pre-peak energy. Based on the experimental analysis, a theoretical model was established, and the surrounding rock stress negatively correlates with the percentage of coal thickness; the energy stored in the surrounding rock is directly proportional to the coal in the zone. Therefore, we inferred that the stress distribution of the surrounding rock as coal thickness changes is abnormal; substantial energy accumulation can swiftly initiate dynamic disasters, such as rock bursts. This study has important reference significance for preventing and controlling rock bursts in areas where coal thickness changes.

**Keywords:** coal thickness change; combination; combined rock strata; energy accumulation; rock burst





## 1. Introduction

With the intensification of coal mining, disasters caused by coal–rock are becoming increasingly severe. Rock bursts pose a significant threat to safe and efficient coal mine production, given its rapid occurrence, high level of harm, and extensive damage caused [1–3]. Rock burst is a dynamic event. It is caused by the sudden release of the elastic energy stored in coal and rock formations. Variations influence this phenomenon in coal thickness and the stress state of the coal seam and rock strata in the stopes [4,5]. Relevant research shows that the probability of rock burst is related to the change in coal thickness. During mining, the energy accumulation in areas with coal thickness change is high, potentially leading to catastrophic events such as rock bursts [6,7].

Rock bursts result from the joint action of coal and rock under the influence of mining. AFRAEI was used to investigate the impact of the mechanical parameters of coal and rock on rock bursts [8]. By analyzing the mechanical rock parameters of many engineering cases, it was established that the rock stress ratio and the elastic strain energy significantly influence the impact of ground pressure. Several scholars have utilized the coal–rock combination to conduct research, and the impact of changes in coal thickness on the strength and

deformation characteristics of coal–rock combinations has been analyzed [9,10]. The results demonstrated that the strength of combinations is negatively correlated with changes in coal thickness. References [11,12] revealed the relationship between coal thickness changes, parameters at the coal–rock interface, and the strength of the combinations. Zuo et al. [13] compared the energy accumulation differences of various coal–rock combinations and identified a correlation with coal thickness variations. Chen et al. [14,15] placed the key layer where energy for rock bursts accumulates within the coal seam, where the degree of energy accumulation was related to the change in coal thickness. Through case analysis, laboratory experiments, and other methods, the above researchers identified the relationship between the combination's strength and energy accumulation behavior characteristics and the mechanical parameters of coal and rock, which has significant reference value for understanding rock bursts. Still, the research on the stress and energy evolution law of surrounding rock in coal thickness change is insufficient.

The change in coal thickness has a particular influence on the stress distribution of the surrounding rock, which affects the energy accumulation characteristics of the surrounding rock. To this end, Bai et al. [16] analyzed the surrounding rock's stress and energy distribution characteristics in an area with roof thickness changes. They concluded that the greater the roof thickness change, the greater the initial stress in the area of coal seam thinning. Wang et al. [17] used the theoretical analysis method to study the stress distribution characteristics of the surrounding rock in an extra-thick coal seam. Refs. [18,19] used numerical simulation to find that abnormal mining stress in the area where coal thickness changes induces rock bursts. Refs. [20,21] used a numerical simulation method to study the elastic strain energy distribution characteristics and influencing factors in an area of changing coal thickness change. Wang et al. [22] pointed out that high-stress concentrations in coal and rock strata increase the possibility of rock bursts. Zhao et al. [23] discussed the mechanical mechanism of mining rock bursts in areas with changing coal thickness. The above scholars have taken the coal–rock combination layer as the object. Through indoor experiments and numerical simulation, they have discussed the combined rock strata's stress and energy distribution laws in areas of changing coal thickness, and it is believed that stress anomalies induce the occurrence of rock bursts. However, the research on the relationship between the change in coal thickness and the stress and energy of the combined rock strata is relatively scarce.

Therefore, this study used the weak impact tendency of the #33 layer coal seam in Jixi Pinggang Coal Mine as the research background. Taking the combined rock strata as the research object, the strata in the area with changing coal thickness were regarded as the change in the coal thickness, and how coal thickness and lithology impact the combination's strength and energy accumulation characteristics was investigated. Based on our analysis, we established a correlation between the stress and energy of the combined rock strata and the subsequent change in coal thickness, enabling us to identify the areas where stress and energy accumulate in the surrounding rock formations as the coal thickness changes. This valuable insight can serve as a guide to effectively prevent and manage rock bursts in the affected areas.

## 2. Uniaxial Compression Experiment of Coal–Rock Combination

To thoroughly examine the combined structure's stress and energy accumulation characteristics within the area with changing coal thickness, the change was regarded as the thickness change of the coal–rock combination. Therefore, in this experiment, different combinations of lithologies and coal thicknesses were made. Then, the uniaxial compression experiment was carried out using an RMT-150 testing machine to obtain the axial stress–axial strain curve of the different combinations.

### 2.1. Research Context

The samples taken in this experiment are from Pinggang Coal Mine in Jixi, Heilongjiang Province. The 33 # coal seam of this coal mine has a weak impact tendency. The

right working face of the 33# coal seam in Pinggang Mine has a depth of 650 m~680 m from the surface, with an average dip angle of 3°. The thickness of the coal seam is 1.1 m~3 m, with an average thickness of 2.1 m. The coal seam roof is fine sandstone with a thickness of 9.6 m, and its strength is high, with a uniaxial compressive strength of 74.5 MPa; the floor is shale with a thickness of 0.3 m.

### 2.2. Samples Making

According to the field conditions, the roof fine sandstone, floor shale, and coal seam of the working face are sampled. The samples are combined according to the experimental requirements, and the combination method is shown in Figure 1a. Among them are fine sandstone (F), shale (S), and coal (C). The size of coal and rock monomer and combination is a cylinder with a diameter of 50 mm and a height of 100 mm. The binder is used in the test to fix the coal–rock interface. To reduce the impact of the coal–rock interface and the inhomogeneity of the combined sample on the test results, the wave velocity of the combination was measured, and the combination with similar wave velocity was selected for the test. Wave velocity measurement is shown in Figure 1b. The physical diagram of some samples is shown in Figure 2.

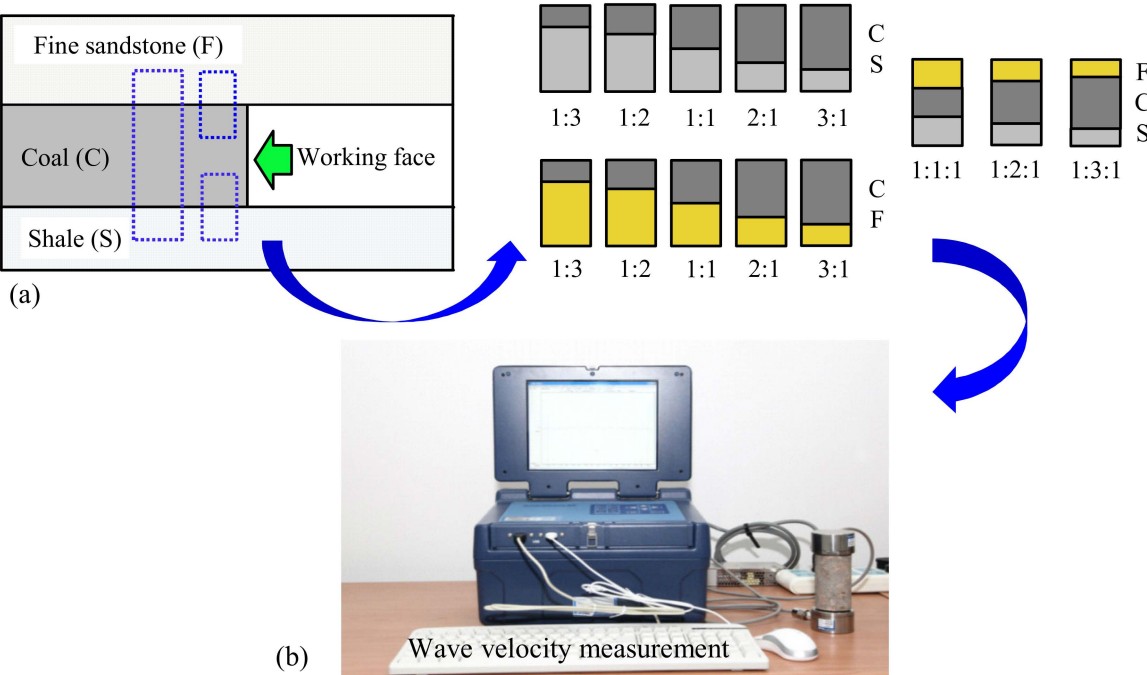

**Figure 1.** Combination model and wave velocity measurement.

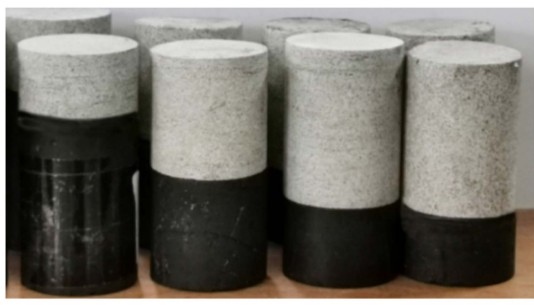

**Figure 2.** Part of the sample physical picture.

### 2.3. Stress–Strain Curve of Coal–Rock Combination

After the sample was made, the 0.05 mm/s loading rate was used to experiment, and the corresponding $\sigma$-$\varepsilon$ curves of F, S, C monomers and CS, CF, and FCS combinations were obtained. The $\sigma$-$\varepsilon$ curves for various structures formed from the combination of coal and rock are displayed in Figure 3. Here, $\sigma$ is the axial stress, and $\varepsilon$ is the axial strain.

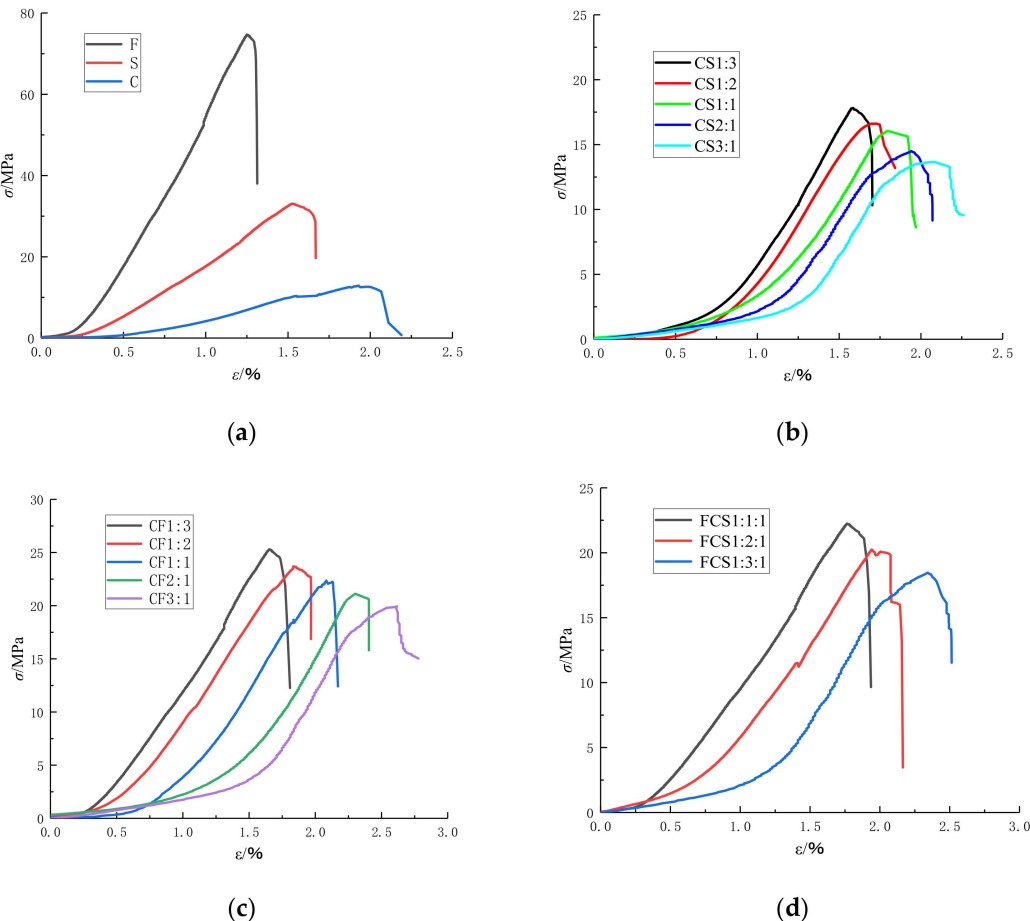

**Figure 3.** $\sigma$-$\varepsilon$ curves of different coal–rock combinations. (**a**) Coal and rock monomer; (**b**) SC combination; (**c**) FC combination; (**d**) FSC combination.

We extracted and compared the mechanical parameters of the monomer and combination in Figure 3 and obtained the mechanical parameter data of the combination, as shown in Table 1.

**Table 1.** Mechanical parameters of the combination.

| Sample | Peak Strength/MPa | Elastic Modulus/GPa | Pre-Peak Energy /$10^2$ MJ·m$^{-3}$ | Post-Peak Energy /$10^2$ MJ·m$^{-3}$ |
|---|---|---|---|---|
| F | 74.50 | 7.48 | 36.9 | 4.15 |
| S | 33.10 | 2.70 | 19.83 | 4.61 |
| C | 12.72 | 0.96 | 9.36 | 2.45 |
| CS1:3 | 17.7 | 2.28 | 8.04 | 2.16 |
| CS1:2 | 16.62 | 1.98 | 8.22 | 2.37 |
| CS1:1 | 16.05 | 1.8 | 8.39 | 2.55 |
| CS2:1 | 14.49 | 1.63 | 8.61 | 2.07 |
| CS3:1 | 13.66 | 1.07 | 8.71 | 2.15 |
| CF1:3 | 25.3 | 2.35 | 15.82 | 3.43 |

**Table 1.** *Cont.*

| Sample | Peak Strength/MPa | Elastic Modulus/GPa | Pre-Peak Energy /$10^2$ MJ·m$^{-3}$ | Post-Peak Energy /$10^2$ MJ·m$^{-3}$ |
|---|---|---|---|---|
| CF1:2 | 23.69 | 2.05 | 16.22 | 2.96 |
| CF1:1 | 22.36 | 1.85 | 16.27 | 2.08 |
| CF2:1 | 21.12 | 1.73 | 15.89 | 2.18 |
| CF3:1 | 19.94 | 1.52 | 17.93 | 2.66 |
| FCS1:1:1 | 22.24 | 1.81 | 14.53 | 3.38 |
| FCS1:2:1 | 20.25 | 1.63 | 14.67 | 3.92 |
| FCS1:3:1 | 18.46 | 1.42 | 14.96 | 2.87 |

## 3. Strength and Elastic Modulus Analysis of Combination

The correlation between the strength and elastic modulus of the coal–rock monomer and combination can be ascertained through the analysis of the mechanical parameters outlined in Table 1, which allows us to examine the effect of the coal thickness and lithology on the strength and modulus of the combination.

### 3.1. Relationship between Strength and Elastic Modulus of Coal–Rock Combination and Coal–Rock Monomer

The peak strength and elastic modulus of various coal–rock combinations are displayed in Figure 4, demonstrating their distribution characteristics. Here, $\sigma_1$ denotes the peak strength, and $E$ represents the elastic modulus.

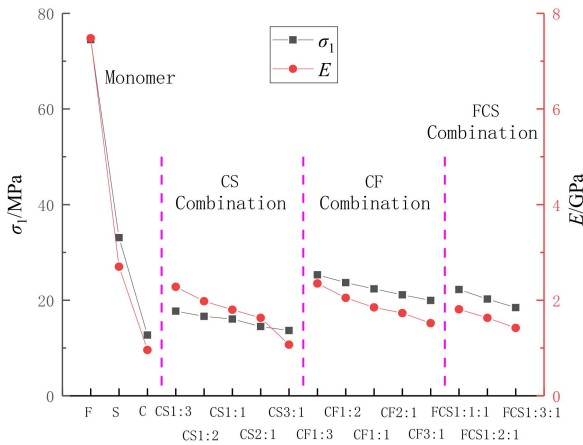

**Figure 4.** Distribution characteristics of peak strength and elastic modulus of different combinations.

Table 1 and Figure 4 demonstrate that the monomer's peak strength and elastic modulus follow the F > S > C order. The binary combination of peak strength and elastic modulus demonstrates that CS is more significant than C and less than S, and CF is more excellent than C and less than F. The ternary combination FCS peak strength and elastic modulus as a whole is more significant than C and less than F and S. This indicates that the coal–rock combination peak strength and elastic modulus are both between the coal and the rock strengths but are closer to the coal monoliths.

### 3.2. Influence of Coal Thickness on the Strength of Combinations

To analyze the impact of the coal thickness on the strength of the combination, we extracted the combination's peak strength and elastic modulus data in Table 1. We established the correlation between the coal–rock combination's peak strength, elastic modulus, and height ratio, as illustrated in Figure 5.

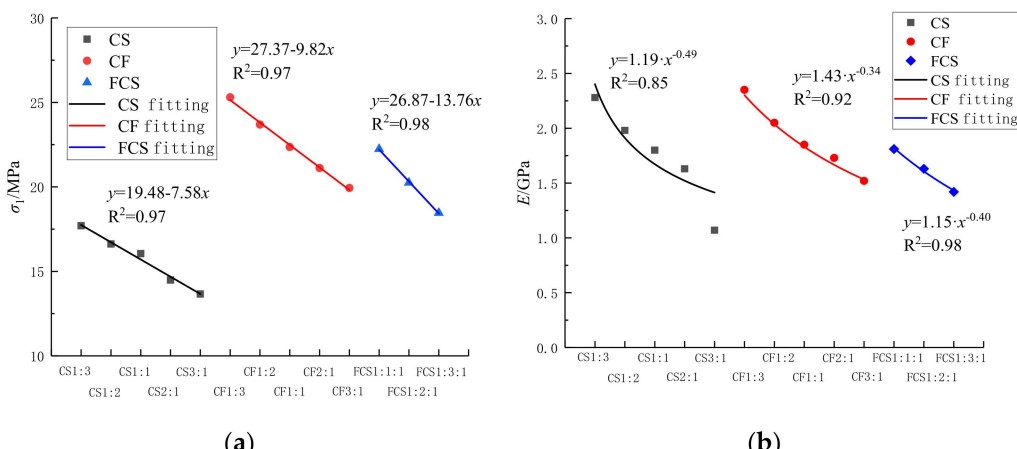

**Figure 5.** The relationship between peak strength, elastic modulus, and coal–rock thickness ratio of the combination. (**a**) Combination $\sigma_1$; (**b**) Combination $E$.

The data presented in Figure 5 show that as the C thickness ratio increases, the peak strength and elastic modulus of CS, CF, and FCS decrease. The peak strength and C thickness of the combination CS, CF, and FCS are represented by the expressions $y = 19.48 - 7.58\,x$, $y = 27.37 - 9.82\,x$, and $y = 26.87 - 13.76\,x$, respectively, and the elastic modulus of the combination CS, CF, and FCS are represented by the ratios of the thicknesses of the combinations to the thicknesses of the C. The CS, CF, and FCS to the thicknesses of the C are represented by the expressions $y = 1.19\,x^{-0.49}$, $y = 1.43\,x^{-0.34}$, and $y = 1.15\,x^{-0.40}$ expressions; this is mainly because the more significant the proportion of C in the combinations, the smaller the axial pressure required for the combinations to undergo axial deformation, and thus the smaller the peak strength and modulus of elasticity of the combinations. Therefore, it can be concluded that the peak strength of the combination is directly linked to the proportion of coal thickness. At the same time, the elastic modulus exhibits an inverse relationship with the ratio of coal thickness.

### 3.3. Influence of Lithology on the Strength of Combinations

From Figure 5, it can be seen that the ratio of CS, CF peak strength, and C thickness of the combination conforms to the expressions of $y = 19.48 - 7.58\,x$ and $y = 27.37 - 9.82\,x$, respectively; the ratio of CS and CF elastic modulus to C thickness of the composite conforms to the expressions of $y = 1.19\,x^{-0.49}$ and $y = 1.43\,x^{-0.34}$, respectively. The variations in peak strength and elastic modulus reduction rate of combinations are primarily due to differing rock strength and elastic modulus of various lithologic combinations. When the coal thickness is constant, the peak strength and elastic modulus of CS and CF combinations are as follows: CF1: 3 > CS1: 3, CF1: 2 > CS1: 2, CF1: 1 > CS1: 1, CF2: 1 > CS2: 1, CF3: 1 > CS3: 1. This demonstrates that the more significant the strength and modulus of elasticity of the rock, the more significant the peak strength and modulus of elasticity of the combination of various combinations of equal coal thickness.

## 4. Energy Accumulation Law of Coal–Rock Combination Body

The patterns of variation in pre-peak energy, post-peak energy, and impact energy index were examined to determine the influence of lithology and coal thickness on the energy accumulation behavior of the combination.

### 4.1. Energy Accumulation Characteristics of Combination Body

The abbreviations used were $A_1$ for pre-peak energy and $A_2$ for post-peak energy. The impact energy index is the ratio of the pre-peak accumulated deformation energy to the post-peak consumed deformation energy of the stress–strain curve of the specimen under

uniaxial compression, which better reflects the energy accumulation and consumption process. It is expressed by $K_E$ and calculated as follows:

$$K_E = \frac{A_1}{A_2} \tag{1}$$

The $A_1$ and $A_2$ data are extracted from Table 1, and then $K_E$ is calculated by Equation (1), leading to the energy distribution law of various combinations. The derived law is illustrated in Figure 6.

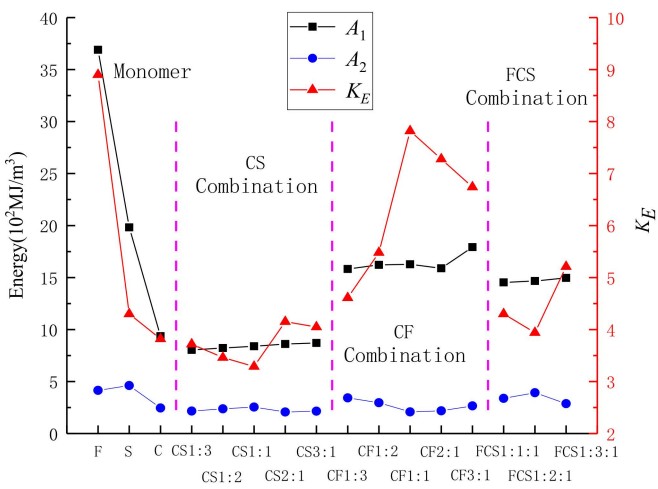

**Figure 6.** Energy variation of different coal–rock combinations.

From the data presented in Figure 6, it is evident that the energy accumulation of the monomers is different from that of the combinations, and the pre-peak energy of the monomers is F > S > C, indicating that the pre-peak energy accumulation of the rock is more significant than that of the coal. The pre-peak energy exhibited by binary combinations suggests that CS exceeds C and is below S, while CF surpasses C and is lower than F. These findings indicate that the lithology significantly impacts the pre-peak energy of the combination. The peak energy of the ternary combination of FCS is greater than that of C but less than that of S and more significant than that of CS. It is slightly less than that of CF, indicating that F has a particular effect on the buildup of pre-peak energy in the ternary combination of FCS. Furthermore, due to F's significant strength and modulus of elasticity compared to S and C, F can be considered a hard rock layer, suggesting that the hard rock layer plays a role in the buildup of pre-peak energy in the combination.

No apparent pattern is discernible in the post-peak energy of the combinations. The energy impact index of the combinations shows that CF is more significant than CS, and FCS is greater than CS, indicating that the greater the rock strength and elastic modulus, the more significant the energy impact index of the combination.

### 4.2. The Influence of Coal Thickness on the Energy Accumulation Characteristics of the Combination

When the coal thickness varies, we extract the relevant data from Table 1 and depict the resulting energy change profile for different combinations of coal thickness, as shown in Figure 7.

Figure 7a demonstrates that the CS and CF pre-peak energy of the binary combinations are arranged in descending order as CS1:3 < CS1:2 < CS1:1 < CS2:1 < CS3:1 and CF1:3 < CF1:2 < CF1:1 < CF2:1 < CF3:1, respectively. In addition, the FCS pre-peak energy of the ternary combination is arranged in descending order as FCS1:1:1 < FCS1:2:1 < FCS1:3.1. This observation suggests a gradual increase in the pre-peak energy of the combination as the percentage of C thickness is augmented.

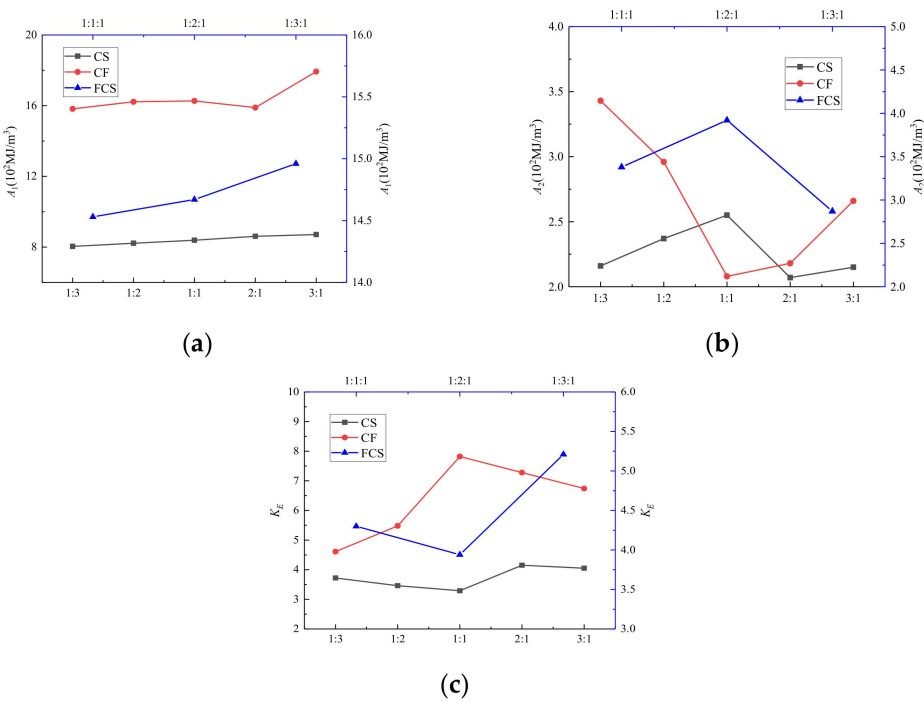

**Figure 7.** The energy variation of the combinations with different coal thicknesss. (**a**) pre-peak energy distribution; (**b**) post-peak energy distribution; (**c**) impact energy index.

As indicated in Figure 7b, as the C thickness percentage increases, the post-peak energy of the combination shows the trend of CS and FCS increasing and then decreasing; on the contrary, CF shows the direction of reducing and then expanding, and the fluctuation range of post-peak energy of the combination CS, CF, and FCS are 2.07~2.55 ($10^2$ MJ·m$^{-3}$), 2.08~3.43 ($10^2$ MJ·m$^{-3}$), and 2.87~3.43 ($10^2$ MJ·m$^{-3}$), respectively. There is no apparent regularity in the post-peak energy of the combination with the more significant coal thickness percentage; the range of post-peak energy fluctuation of the CF combination is more extensive than that of the CS combination, which shows that the larger the rock strength and elastic modulus are, the larger the range of post-peak energy fluctuation of the combination.

According to Figure 7c, binary combination CS and CF exhibit a fluctuation range of impact energy index between 3.29 and 4.15 and 4.61 and 7.82, respectively. Additionally, ternary combination FCS shows a fluctuation range of impact energy index between 3.94 and 5. 21. The impact energy index of combination is affected by the lithology, with greater peak strength and modulus of elasticity in rocks resulting in the broader fluctuation range. The lithology affects the combination's impact energy index, with greater peak strength and modulus of elasticity in rocks, resulting in a broader fluctuation range. Generally, an increasing trend in the impact energy index of the combination is observed with a more significant proportion of coal thickness.

The results suggest that a higher percentage of coal thickness in the combination leads to an increase in the pre-peak energy accumulation. The post-peak energy displays no clear patterns, while the impact energy index indicates a rising trend.

### 4.3. The Influence of Lithology on the Energy Accumulation Characteristics of the Combination

We extracted energy data from CS and CF combinations of equal coal thickness and plotted different combination energy change curves (Figure 8).

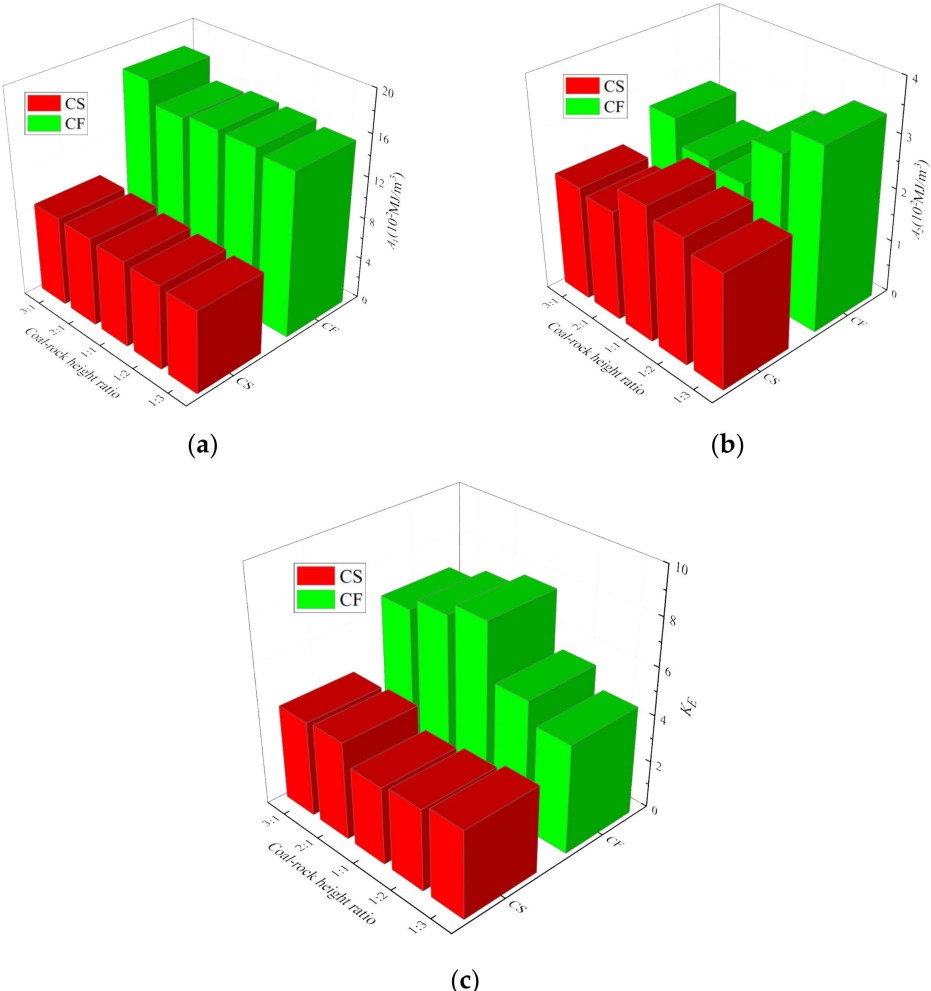

**Figure 8.** Energy variation of different lithologic combinations. (**a**) Average pre-peak energy; (**b**) Average post-peak energy; (**c**) Impact energy index.

When the coal thickness is constant, Figure 8a shows that the pre-peak energy of CS and CF vary as follows: CF1:3 > CS1:3, CF1:2 > CS1:2, CF1:1 > CS1:1, CF2:1 > CS2:1, and CF3:1 > CS3:1. In addition, the pre-peak energy of the CF combination is more significant than that of the CS combination. As the F peak strength and elastic modulus are more significant than the S, the more significant the peak rock strength and elastic modulus, the greater the pre-peak energy buildup of the combination.

Figure 8b shows that the post-peak energy of the combination CS fluctuates in the range of 2.07~2.55 ($10^2$ MJ $\cdot$ m$^{-3}$). The CF post-peak energy of the combination fluctuates in the range of 2.08~3.43 ($10^2$ MJ $\cdot$ m$^{-3}$). The post-peak energy fluctuation range of the CS combination is significantly smaller than that of the CF combination. It shows that the larger the rock monomer's peak strength and elastic modulus, the more extensive the post-peak energy fluctuation range of the combination.

As shown in Figure 8c, when the coal thickness is equal, the CS and CF impact energy index of the combinations are CF1:3 > CS1:3, CF1:2 > CS1:2, CF1:1 > CF1:1, CF2:1 > CS2:1 and CF3:1 > CS3:1. The fluctuation range of the CS and CF impact energy index of the assemblage is 3.29~4.15, respectively, 4.61~7.82. In addition, the rock's peak strength and elastic modulus directly affect the impact energy index, with larger values suggesting a higher impact potential.

## 5. Coal–Rock Combination Structure Model

### 5.1. Mechanical Model of Coal–Rock Combination Structure

Under the assumption that the coal–rock structure is in an elastic state before the damage occurs, the coal–rock structure is represented as a spring structure consisting of two elastic elements connected in series [15], as the model shown in Figure 9.

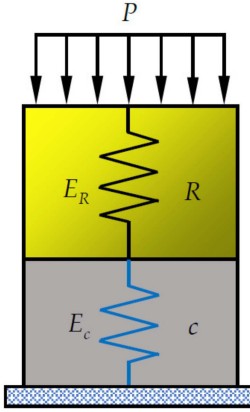

**Figure 9.** Coal–rock combination model.

Under mining stress P, coal and rock are subject to deformation to a certain extent, while their stress distribution is equivalent. Assuming that the overall elastic modulus of the combination is E, the height of coal and rock is $h_c$ and $h_R$, respectively, the deformation is $\Delta h_c$ and $\Delta h_R$, respectively, and the elastic modulus is $E_c$ and $E_R$, respectively.

$$P = E_R \varepsilon_R = E_c \varepsilon_c \tag{2}$$

$$\varepsilon_c = \frac{\Delta h_c}{h_c} \quad \varepsilon_R = \frac{\Delta h_R}{h_R} \tag{3}$$

$$\varepsilon = \varepsilon_c + \varepsilon_R = \frac{p}{E} \tag{4}$$

$$\frac{1}{E} = \frac{1}{E_R} + \left( \frac{1}{E_c} - \frac{1}{E_R} \right) \frac{h_c}{H} \tag{5}$$

Equation (4) can be obtained by substituting Equations (2) and (3) into Equation (4). It can be seen that when the total height of the coal–rock combination remains unchanged, and the elastic modulus of coal and rock remains intact, the overall elastic modulus of the combinations decreases with the increase of coal thickness $hc$.

### 5.2. Relationship between Combinations of Stress and Coal Thickness

Assuming that the same strain $\varepsilon$ occurs after the coal and coal–rock structures are stressed, it can be seen from the elasticity [24]:

$$P_c = E_c \varepsilon \tag{6}$$

$$P_{cR} = E \varepsilon \tag{7}$$

The stress of coal–rock structure is compared with that of coal [16,25], that is

$$\frac{P_c}{P_{cR}} = \frac{E_c}{E_R} + \left( 1 - \frac{E_c}{E_R} \right) \frac{h_c}{H} \tag{8}$$

As can be seen from Equation (8), $E_c < E_R$, $E_c/E_R < 1$, $h_c/H < 1$, $E_c/E_R + (1-E_c/E_R) h_c/H < 1$, that is, $P_c < P_{cR}$. When the coal and rock strain is the same, the stress on the coal–rock structure is more significant than that on coal. Meanwhile, the stress of surrounding rock in the coal thickness thinning region is more significant than that in the coal thickness

changing area, and the more significant the proportion of coal thickness, the smaller the stress in the coal thickness changing region.

### 5.3. Relationship between Combinations of Energy and Coal Thickness

There is a commonly held belief that the coal body is destroyed when mining stress reaches its maximum, and the stress of the coal–rock combination reaches its peak. The minimum energy theory of rock mass failure posits that the energy consumed by coal and rock mass failure is the energy under a unidirectional stress state. The energy expended during coal and rock mass failure is typically associated with a unidirectional stress state [26–28].

$$U = \frac{P_{max}^2}{2E} = \frac{P_{max}^2}{2} \times \left[ \frac{1}{E_R} + \left( \frac{1}{E_c} - \frac{1}{E_R} \right) \frac{h_c}{H} \right] \tag{9}$$

From the laboratory determination of mechanical parameters and the field test of Pinggang Coal Mine and the results, the elastic modulus of coal, fine sandstone, and shale are 0.96 GPa, 7.48 GPa, and 2.70 GPa, respectively. The stress concentration coefficients under the influence of mining are 2~2.5, corresponding to stresses of 35.0 MPa and 43.75 MPa. The graph in Figure 10 demonstrates the correlation between energy accumulation and coal thickness.

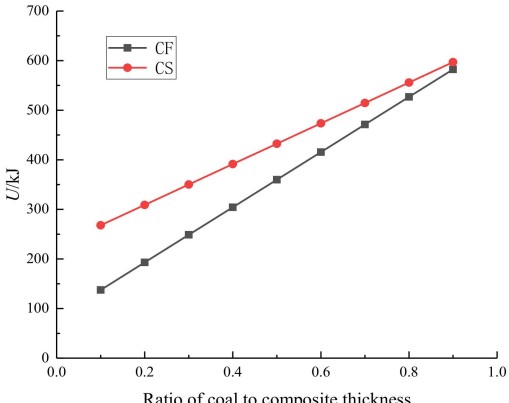

**Figure 10.** Relationship between the energy of coal–rock combinations and the ratio of coal thickness.

Equation (9) and Figure 10 show that for constant coal and rock elastic modulus, the energy accumulated before the destruction of the formation is directly related to the coal thickness hc. In the case of continuous coal and rock thickness, the greater the modulus of elasticity of the rock, the more energy is accumulated and the greater the possibility of impact hazard. These observations corroborate the results of the experiment.

## 6. Discussion

In coal mining, rock bursts occur due to the stress state of the coal seam and rock stratum and their occurrence conditions. The thickness of the coal seam varies significantly in the local area, and the stress state of the surrounding rock in this area is correlated with the variation in coal thickness and the mechanical properties of the roof and floor.

The above research (Section 4.2) shows that the surrounding rock stress is negatively correlated with the percentage of coal thickness. The surrounding rock stress is more significant in the area of thinning coal thickness than in the area of thickening, which indicates that during the process of the coal seam changing from thick to thin, the degree of concentration of surrounding rock stress in the area of coal thickness change gradually increases, which is easily induced the rock burst.

The law of energy accumulation in combined strata is influenced by coal thickness, rock strength, and elastic modulus. A higher proportion of coal thickness increases energy accumulation before the combination peak. A thick coal seam area accumulates more energy than a thinner one and thus is more likely to experience rock bursts. The positive

correlation between the combined rock strata's energy accumulation and the rock's strength and elastic modulus indicates that the bigger the strength of the roof and floor, the more the combined rock strata's energy accumulation is, and the more prone to rock burst.

In summary, the stress distribution in the area where the thickness of coal changes shows abnormality. There is a significant stress concentration in the region where the coal thickness is thinning. The energy accumulation in thick coal seams is higher than in thin coal seams. Consequently, the surrounding rock where the thickness of the coal changes and in the thick coal seam area is more likely to accumulate energy, leading to dynamic disasters such as rock bursts, as illustrated in Figure 11.

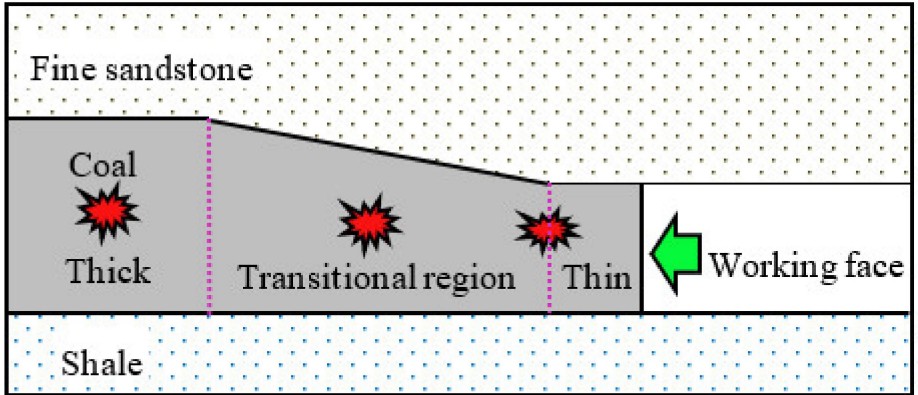

**Figure 11.** Coal thickness change model.

In this paper, the coal–rock combination is considered a binder, and the coal–rock interface is regarded as a uniform medium, without considering the influence of the coal–rock interface properties on the energy storage and release of the combination. Therefore, the following work will focus on the coal–rock interface properties and the impact of different binders on the energy storage and release of the combination.

## 7. Conclusions

Rock bursts result from the joint action of coal and rock under the influence of mining. To study the impact of the mechanical properties of coal–rock on rock burst, this paper takes the coal–rock combination as the object, analyzes the effects of different coal thickness and lithology on the mechanical parameters and energy accumulation behavior of the combination, reveals the change of coal thickness and the combination of rock layer stress and energy between, and reaches the following conclusions:

1. The coal–rock combinations' peak strength, elastic modulus, and pre-peak energy are between coal and rock. In contrast, the post-peak energy and impact energy indexes show no apparent regularities with coal and rock;

2. As the coal thickness ratio increases, the peak strength and elastic modulus of combinations gradually decrease, except for strength, which has a clear linear relationship to the coal thickness ratio, and elastic modulus, which has a clear inverse relationship to the coal thickness ratio. The combination's pre-peak energy and impact energy index shows an apparent positive correlation with coal thickness, while post-peak energy shows no obvious regular pattern;

3. Coal thickness accounted for the same percentage; the combination's strength and modulus of elasticity positively correlate with the rock's strength and elastic modulus. The pre-peak energy of the combination, impact energy index, and rock strength and modulus of elasticity showed a positive correlation, and the hard rock stratum helps the combination pre-peak energy accumulation, the greater the likelihood of its occurrence of impact;

4. The mechanical model of the coal–rock combination is established. The surrounding rock stress shows a negative correlation with the percentage of coal thickness, and the

surrounding rock energy shows a positive correlation with the rate of coal thickness, indicating that surrounding rock stress is more significant in the thinning coal thickness region than in the thickening coal thickness region and surrounding rock energy accumulation is less in the thinning coal thickness region than in the thickening coal thickness region. The abnormal stress distribution and high energy accumulation of surrounding rock in the area of coal thickness change easily induce dynamic disasters such as rock bursts.

**Author Contributions:** Conceptualization, G.Z.; validation, Z.L.; formal analysis and investigation, Z.L.; data curation, Z.L., Y.L. and W.Z.; writing—original draft preparation, Z.L.; writing—review and editing, Z.L. and Y.L.; supervision, G.L., T.Q. and L.Z. All authors have read and agreed to the published version of the manuscript.

**Funding:** This work was supported by the Scientific and Technological Key Project of "Revealing the List and Taking Command" in Heilongjiang Province (2021ZXJ02A03, 2021ZXJ02A04) and the National Natural Science Foundation (51774122).

**Institutional Review Board Statement:** Not applicable.

**Informed Consent Statement:** Not applicable.

**Data Availability Statement:** Not applicable.

**Conflicts of Interest:** The authors declare no conflict of interest.

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
