# Peer review of "Energy Accumulation Law of Different Forms of Coal–Rock Combinations"

_applsci, doi:10.3390/app132011393_

Round 1
Reviewer 1 Report
The paper gives a scientific methodolgy to evaluate the energy (strain energy) storage and release by physical models and laboratory invetigation. However, it is not obvious that how the combination models were built and what would be the impact of interfaces between coal, sandstone and shale on the stored energy and its release. Also, it has not been described as to how uniformity in the binding process could be achieved and ascertained. Furthermore, the paper must present the method of validating the observations and results.
Minor errors in English language and grammar have been observed. These must be cortected.
Author Response
Add the model building scenario in Figure 1 (a). In this paper, the coal-rock interface adopts the binder mode, and the influence of coal-rock interface on energy storage and release is the focus of the next study, which has been added in the discussion. In order to achieve the uniformity of binding samples, the method of wave velocity test can be adopted, as shown in Figure 1 (b).
Reviewer 2 Report
The manuscript "Energy accumulation law of different forms of coal-rock combi-nations" by Li Zibo, Zhang Guohua, Li Yubo, Zhou Wenjun, Qin Tao and Zeng Li ,Liu Gang was submitted for review.
The manuscript has a number of significant deficiencies that need to be corrected. Correction of the following deficiencies is necessary to improve the quality of the manuscript, enhance the ease of comprehension of the material presented, and increase the interest of the reader.
1) The abstract is not properly formed. Why did the authors copy the conclusions? The abstract should clearly indicate the purpose of the study, its importance for society (i.e. characterize the problem), indicate the methods and materials of the study, and the conclusions should be clearly and briefly formulated. The abstract lacks a "starting point", i.e. information about previously conducted research (one sentence is sufficient).
The abstract should be redone.
2) The authors did not summarize their analysis in the introduction. When conducting a literature review, the authors relied only on the research of scientists of the Chinese scientific school, which does not fully characterize the actual question posed, the purpose of the study and the tasks to be solved to achieve this goal. For example: "Analyzing the above it can be noted that ......" is a very relevant issue. Consequently, the aim of this study is ..... and to achieve which the following tasks should be solved:1); 2); ......
Such a conclusion at the end allows the researchers to properly formulate the conclusions of the study and the reader to understand the vector of the study.
Needs to be finalized.
3) From my point of view, the authors misuse the surnames of scientists when mentioning the research. The authors indicate the name of the researcher (or group of researchers) then indicate their achievement, followed by a reference to the study. In my opinion, a reference without mentioning the last name at the beginning of the sentence is sufficient. If the reader will be interested to know the name of the researcher, he/she will refer to the reference. The reader is interested in the essence of the issue being disclosed, not the surname of the researcher who disclosed the issue.
4) Group references should be avoided. Each paper you cite is unique and the research you cite deserves more appropriate and careful review to demonstrate (and prove) its importance to the current research. You should demonstrate in detail the essence of each study and their necessity in your work (Example: Reference [ 17-20 ] examines the impact....). No more than 2 references are allowed.
5) Why did the authors choose the 33 # layer coal seam at the Jixi Pinggang coal mine as the object of their study. Why the manuscript does not reflect the mining and geological conditions of the coal seam under consideration (coal height, bedding angle, rock strength, etc.)?
6) The authors need to explain what Impact energy index is and how it is calculated?
7) In conclusion, the authors of the manuscript should conceptualize the purpose of the study. Conclusion is the summary of the study done by the authors without repetition. Conclusions should briefly characterize the result of the study, e.g.: as a result of the study a dependence was obtained; established, and so on. The authors should support all their conclusions with numerical evidence.
8) The manuscript has a not sufficient reference list (26 references in total). However, there is a lack of comprehensive coverage of research in terms of citation geography. There are no references to the world experience of research in the field or related fields, especially to the works of Eastern European, Ukrainian or Russian scientists, for example:
- Gabov, V.V., Zadkov, D.A., Babyr, N.V., Xie, F. (2021). Nonimpact rock pressure regulation with energy recovery into the hydraulic system of the longwall powered support. Eurasian Mining, 36(2), 55-59. doi:10.17580/em.2021.02.12.
- Gendler, S.G., Gabov, V.V., Babyr, N.V., Prokhorova, E.A. (2022). Justification of engineering solutions on reduction of occupational traumatism in coal longwalls. Mining Informational and Analytical Bulletin, (1), 5-19. doi:10.25018/0236_1493_2022_1_0_5.
As follows from the presented works, the authors of the manuscript submitted for review missed a rather large layer of research. If the authors of the manuscript submitted for review familiarize themselves with the presented works, they will be able to properly form the introduction and enrich their manuscript with international studies of scientists.
It is mandatory to supplement the reference list with studies of scientists from different countries in the last 3-5 years to show geographical (general/world) interest and relevance.
9) Additional remarks.
- There are spelling errors in the manuscript. Example: there are spaces before references [ ].
- first mentioning the figures in the text, then the figures themselves. Example: figure 3.
Moderate editing of English language required
Author Response
(1)The format of the abstract has been revised and the purpose of the study and its importance to society have been added, as detailed in the abstract.
(2)The results of relevant national researchers have been added to the introduction, and the introduction has been revised, please refer to the introduction for details.
(3)The format of the references has been revised and the author's last name has been removed, leaving only the author. See introduction for details.
(4)The format of references has been revised, see introduction for details.
(5)Add engineering background in Article 2.1.The 33 # coal seam of this coal mine has a weak impact tendency.he 33# coal seam in Pinggang Mine has a depth of 650m~680m from the surface, with an average dip angle of 3°. The thickness of the coal seam is 1.1m~3m, with an average thickness of 2.1m. The coal seam roof is fine sandstone with a thickness of 9.6m, and its strength is high, with a uniaxial compressive strength of 74.5MPa; the floor is shale with a thickness of 0.3m.
(6)The impact energy index is the ratio of the pre-peak accumulated deformation energy to the post-peak consumed deformation energy of the stress-strain curve of the specimen under uniaxial compression, which better reflects the energy accumulation and consumption process. It is expressed by KE and calculated as follows :KE=A1/A2
(7)The research purpose has been added to the conclusion of the manuscript, and the conclusion is further summarized, see the conclusion for details.
(8)A total of 28 relevant international documents have been added to the manuscript, please refer to the references for details.
(9)The spelling errors in the manuscript have been corrected, and the diagrams have been revised. See article for details.
Round 2
Reviewer 2 Report
The manuscript is a completed research paper. The chosen research topic is indeed relevant. After correcting the comments, the authors were able to present their research correctly and distinctly, which reflects its value and ease of perception.
I thank the authors for the work done and from my point of view, the manuscript can be published in the open access without correct.
Moderate editing of English language required